# OpenReview forum: "Agentic reinforcement learning for search is unsafe"
_ICLR.cc/2026/Conference — Submitted to ICLR 2026_

### Official Review · Reviewer_9TKw · 2025-10-30

**Soundness:** 3
**Presentation:** 2
**Contribution:** 3
**Rating:** 4
**Confidence:** 3

**Summary:**

This paper researches the safety aspect of agentic reinforcement learning. The authors first train the LLM agents from Qwen2.5-7B models and Llama-3.2.3B models through RL and conduct extensive experiments. They conduct extensive studies on both search attacks and multi-search attacks. They find that both attacks degrade the performance, while the agents trained from the IT models are more resistant to attacks. They also find that one harmful search is enough to jailbreak, and iterative searches can lead to more harm.

**Strengths:**

1. The paper researches an interesting problem of whether the model trained from agentic RL is safe or not.
2. The authors conduct extensive experiments to research the problem from an empirical perspective and provide insights.
3. The paper is quite novel.

**Weaknesses:**

1. My main concern with this paper is whether the conclusion from the paper can be generalized.
- Firstly, only small-sized LLMs are approached in this paper. It is quite possible that larger models, even the pretrained checkpoint, are better at preventing jailbreaking and thus perform better with regard to attacks.
- Secondly, the experiments are only conducted on Qwen2.5 and Llama3.2 models. It is possible that these two types of models are not trained with a large amount of safety data during pretraining.

2. I feel that the conclusion drawn in the paper largely depends on what LLMs are used for experiments, and their behaviors are determined by the training data they saw during pretraining.

**Questions:**

1. Do you think the conclusions from the paper can be generalized to larger model sizes and different types of LLMs?
2. I am not sure if the research is valid in this paper since we can always alleviate the problem by adding more safety-related data during pretraining or after RL to alleviate?
3. Is it true that the research conclusions drawn from the paper are customized for Qwen2.5 and Llama3.2, which are determined by the training data for these two types of models, and are hard to generalize?

---

> ### Author Response · Authors · 2025-11-22
> **Correct a Small Misunderstanding and Show Larger-Model Generalisation**
>
> Thank you for your time taken to review our paper. We appreciate your recognition of the novelty of our work. Below we address the concerns you raised.
>
> ---
>
> We start by clarifying a small misunderstanding in your summary. You wrote:
>
> > They find that both attacks degrade the performance, while the agents trained from the IT models are more resistant to attacks.
> >
>
> To clarify: in our paper, **all** jailbreak attacks are applied to RL-trained Instruction-tuned (IT) models, which we refer to as *IT-search* models (Table 3). IT models are therefore *not* more resistant to attacks, they are the only ones being attacked and shown to be vulnerable.
>
> We **did not** evaluate attacks on RL-trained base models (*Base-search* models) as base models (Qwen and Llama) do not refuse harmful instructions even after RL. As in Appendix Figure 10, Base-search Qwen directly followed harmful instructions and generated series of harmful search queries. This shows that these models require no jailbreaks, they are already unsafe by default.
>
> ---
>
> Now we address each of your concerns:
>
> > **W1.1:** My main concern with this paper is whether the conclusion from the paper can be generalized.
> Firstly, only small-sized LLMs are approached in this paper. It is quite possible that larger models, even the pretrained checkpoint, are better at preventing jailbreaking and thus perform better with regard to attacks.
> >
>
> To address your concern concretely, we extended our experiments to **larger Qwen-2.5 models (14B and 32B),** the model family used in over 90% of recent RL for Tool-Integrated Reasoning (RL-TIL) papers ([1-4], ~10 papers in Appendix A), and trained them with both PPO and GRPO.
>
> **Result:** We observed the **same** unsafe-search behaviours and high attack success rates.
>
> The full results are provided in Point 1 of our response, *“To All Reviewers – Changes Added to Paper,”* and Table 4 of the updated PDF.
>
> With these additional results, our attack successes now **generalise** across:
>
> - **Two model families:** Qwen and Llama
> - **Four model scales:** 3B, 7B, 14B, 32B
> - **Two RL algorithms:** PPO and GRPO
> - **Two search setups:** local and web
>
> Existing studies also show that larger open-weight models outside the two families, such as DeepSeek-R1 (671B) and Gemma-3 (9B–27B), remain highly jailbreakable in their base and IT forms [5,6]. This suggests scale alone does not reliably improve safety.
>
> ---
>
> [1] S. Zhu, W. Lin, Y. Zhang; *Search-R1: Training LLMs to Reason and Leverage Search Engines with Reinforcement Learning*, 2025. arXiv:2501.12561.
>
> [2] H. Zhao, Z. Liang, X. Wang; *R1-Searcher: Incentivizing the Search Capability in LLMs via Reinforcement Learning*, 2025. arXiv:2502.08987.
>
> [3] Y. Ji, J. Wang, C. Hu; *Agentic Reinforced Policy Optimization (ARPO)*, 2025. arXiv:2501.13693.
>
> [4] W. Wang, X. Shi, W. Jiao; *ReSearch: Learning to Reason with Search for LLMs via Reinforcement Learning*, 2025. arXiv:2502.19457.
>
> [5] W. Zhang, X. Lei, Z. Liu, et al.; Safety Evaluation of DeepSeek Models in Chinese Contexts, 2025. arXiv:2502.11137.
>
> [6] Promptfoo; Gemma 3 Security Report: Comprehensive AI Red Teaming, 2025.

---

> ### Author Response · Authors · 2025-11-22
> **Why the Vulnerability Does Not Depend on Model Type**
>
> We continue addressing your concerns: whether our attack success is model-specific and dependent on pre-training data.
>
> > **W1.2:** Secondly, the experiments are only conducted on Qwen2.5 and Llama3.2 models. It is possible that these two types of models are not trained with a large amount of safety data during pretraining.
> >
>
> and
>
> > **W2:** I feel that the conclusion drawn in the paper largely depends on what LLMs are used for experiments, and their behaviors are determined by the training data they saw during pretraining.
> >
>
> and
>
> > **Q3:** Is it true that the research conclusions drawn from the paper are customized for Qwen2.5 and Llama3.2, which are determined by the training data for these two types of models, and are hard to generalize?
> >
>
> ---
>
> Thank you for raising these questions. You are essentially asking:
>
> (1) whether our attack success is specific to Qwen2.5 and Llama3.2, and
>
> (2) whether the amount of safety data seen in pre-training determines the model’s behaviour in post-training stages of RL.
>
> These are exactly the kinds of questions we want the community to ask in light of our findings.
>
> To restate, our work focuses on the **relative** safety change **before vs. after RL**, rather than the absolute safety level of pre-training. We revealed a competing objective between post-training stages: standard RL search training (optimised for answer accuracy) can easily conflict with IT (before it can refuse), leading models to produce **unsafe searches**.
>
> Figure 4 shows that these unsafe searches do not come from IT; they arise from the RL training process. This shows that RL opens up **new attack surfaces** that IT was not designed to defend against, a phenomenon not revealed in previous work.
>
> With the contributions clarified, we answer your questions:
>
> ---
>
> ### (1) Is our attack success specific to Qwen2.5 and Llama3.2?
>
> **No, we do not think that is the case.**
>
> As shown in Figure 4 and Discussion, the unsafe search behaviour arises from the **RL training process,** driven by the RL objective design: standard outcome-based rewards in RL motivates the model to emit searches mirroring the harmful request, regardless of whether the query is harmful.  ****This is more of a problem with RL rewards and training data, rather than a property of a particular model family.
>
> Our model choices of these two families are also not arbitrarily. Qwen and Llama are the  SOTA backbones in nearly all recent RL-TIR (tool-integrated reasoning) papers released in 2025, including Search-R1 [1], R1-Search [2], ARPO [3], and ReSearch [4]. We showed in Appendix A, among the top **~10** works released this year (by citations and GitHub stars), over **90%** use Qwen-2.5 and over **40%** use Llama-3.x as the base models for RL search training. These are precisely the models that the open-source community fine-tunes with RL for search-based reasoning today, making them most relevant choices for uncovering safety vulnerabilities during RL.

---

> ### Author Response · Authors · 2025-11-22
> **Why the Vulnerability Does Not Depend on Pre-training Data**
>
> ### (2) Is our attack success determined by the pre-training data?
>
> **No.** Our attacks would still succeed even if pre-training data were perfectly safe, for three reasons:
>
> 1. Unsafe searches as a tool-calling ability is not filtered in pre-trained models;
> 2. Even if they are filtered, RL can still incentivise them by replicating user requests;
> 3. Harmful outputs can come from retrieval, not from the model’s internal knowledge.
>
> ---
>
> 1. **Unsafe searches are not filtered in current pre-trained models**
>
> Public reports for Qwen 2.5, Llama 3, and Gemma 3 [7–9] state that some safety filtering is applied during pre-training (e.g., removal of explicit harmful content). However, to our knowledge, no major open-source model family publicly documents the removal of **tool-use-related** unsafe trajectories during pre-training. As a result, unsafe search behaviours, the primary driver of our attack success, are likely to persist in pre-trained models and can be re-elicited through jailbreaks.
>
> ---
>
> 2. **RL can incentivise unsafe searches by replicating user requests**
>
> If the pre-trained model had never seen any harmful data, it would have no internal representation of what “harm” means to counteract RL’s pressure to mirror the “harmful” request as a search query to maximise QA accuracy. So unsafe searches will still be produced. This is consistent with prior findings that LLMs can generate harmful continuations (e.g., toxicity [10]) even without being explicitly trained on harmful data. We already showed that IT cannot defend against this RL pressure (Figure 4).
>
> ---
>
> 3. **Harmful outputs can come from retrieval, not from the model’s internal knowledge**
>
> For search agents, harmful outputs can stem entirely from retrieved content, rather than model pre-training knowledge. We observed this extensively: Qwen’s reasoning traces often begin with “Based on the information provided…” (Appendix Figure 11), showing its answers rely heavily on retrieved content. In our Multi-search attack, unsafe searches retrieve multiple rounds of harmful content, which progressively shifts model outputs to be harmful (Appendix Figure 12). This functions like a many-shot jailbreak [11], which rely on externally retrieved harm rather than model’s pre-trained harm.
>
> ---
>
> To summarise, our attack success reveals a systematic weakness rooted in the RL training process and its conflict between IT, rather than an artefact of any particular model family. The vulnerability does not depend on pre-training data, as tool-use trajectories are not filtered in pre-training, RL can easily incentivises models to replicate harmful user request, and harm can arise from retrieval and not model pre-training. We hope these answers adequately address your concerns.
>
> ---
>
> [7] Qwen Team. Qwen2.5 Technical Report. arXiv preprint arXiv:2501.10696, 2025.
>
> [8] Meta AI. *The Llama 3 Herd of Models: Technical Report and Model Card.* Meta AI Research Systems Report, 2024.
>
> [9] Google DeepMind. *Gemma 3: Open Models for Responsible AI.* Technical Report, 2025.
>
> [10] A. Gehman, S. Gururangan, M. Sap et al., *RealToxicityPrompts: Evaluating Neural Toxic Degeneration in Language Models.* Findings of the Association for Computational Linguistics (ACL Findings), 2020.
>
> [11] C. Anil, E. Durmus, N. Panickssery, et al.; Many-shot Jailbreaking, 2024. Anthropic Research.

---

> ### Author Response · Authors · 2025-11-22
> **Restate Result Generalisation and Research Validity**
>
> Finally, we address your last set of concerns:
>
> ---
>
> > **Q1:** Do you think the conclusions from the paper can be generalized to larger model sizes and different types of LLMs?
> >
>
> Yes. For generalisation, we now demonstrate attack success across two widely used model families (Qwen and Llama), four model sizes (3B → 32B), two RL algorithms (PPO and GRPO), and two search setups (local and web).
>
> As explained above, the vulnerability we identify is a systematic weakness rooted in RL training process and the conflict between RL and IT, rather than an artefact of any particular model family. This is why we observe the unsafe-search behaviour consistently across different model families, scales, and RL algorithms.
>
> ---
>
> > **Q2:** I am not sure if the research is valid in this paper since we can always alleviate the problem by adding more safety-related data during pretraining or after RL to alleviate?
> >
>
> Thank you for raising this point. Yes, there are likely many ways to alleviate the problem (e.g., modifying the RL reward, adding more safety-aware data in RL) as we summarised in Discussion.
>
> But as with any jailbreak paper, diagnosing the vulnerability is a necessary **prerequisite** for discussing how to mitigate it. In that sense, whether we can later “alleviate the problem” is **orthogonal** to our core contribution: we first need to establish that the problem exists, *where* it comes from, and *which stage* of training is responsible, before we discuss any solutions.
>
> Our findings show that **RL-induced unsafe searches introduce new attack surfaces that IT was not designed to defend against.** This matters because many practitioners assume that IT alone ensures safety, therefore apply outcome-based RL for search without any safety considerations [1-4]. Our paper shows this assumption is incorrect, highlighting the need for practitioners to account for the safety consequences of RL when training search-capable agents.
>
> ---
>
> In summary, we demonstrated that our attack success generalises to larger and stronger models, and clarified why the identified vulnerability is neither model-specific nor dependent on pre-training data.
>
> If our clarifications have addressed your concerns, we kindly ask you to consider raising the score accordingly.

---

> ### Author Response · Authors · 2025-11-27
> **Could you respond to our rebuttal?**
>
> Dear Reviewer,
>
> I hope this message finds you well. As the discussion period is nearing its end with less than three working days remaining, I wanted to ensure we have addressed all your concerns satisfactorily. Could you kindly review our rebuttal and give us some feedback? If there are any additional points you'd like us to consider, please let us know.
>
> Your insights are invaluable to us, and we are eager to address any remaining issues to improve our work. Thank you for your effort in reviewing our paper.

---

### Official Review · Reviewer_bhvc · 2025-11-01

**Soundness:** 2
**Presentation:** 2
**Contribution:** 2
**Rating:** 2
**Confidence:** 4

**Summary:**

The paper examines safety in agentic RL for search: LLMs trained with PPO to alternate between reasoning and tool use (“think → search → answer”) over a local Wikipedia index or web search. Using Qwen-2.5-7B and Llama-3.2-3B (base and instruction-tuned), it optimizes exact-match on HotpotQA and then evaluates 299 harmful prompts. Safety is judged by Prometheus-7B-v2 on refusal, answer safety, and search safety (scaled 0–100). Two simple interventions—forcing an initial search or many searches—significantly erode refusal and safety compared with instruction-tuned baselines.

**Strengths:**

- Safety of RL-trained tool-use agents (especially search) is under-examined and relevant for deployment.

- The three-metric evaluation (refusal, answer safety, search safety) provides nuanced assessment. In addition, findings hold across two model families, both local and web search, demonstrating some generalizability.

- Prefill/prompt tweaks are realistic (user-accessible) and isolate a when-to-search failure mode.

**Weaknesses:**

- Novelty & Impact. The core finding feels incremental and closely aligned with well-known RAG/jailbreak dynamics: if you can steer retrieval early, you can bias generation toward unsafe outcomes.

- Usefulness of the Study: The attack surface here (forcing <search> / multi-search) is quite simple, and the experiments use relatively not strong, non-SOTA models. As a result, it’s unclear whether the phenomenon meaningfully persists, and to what degree, in production-grade systems (e.g., Claude, Gemini, ChatGPT) that already deploy safety scaffolds and gated tool use. Without evidence on stronger models or more realistic threat models, I’m not convinced the contribution is broadly useful.

- The RL setup (PPO on HotpotQA with tool tokens) is quite specific and not clearly representative of how real systems are trained/deployed today.

- RL reward optimizes exact match on HotpotQA, but safety is evaluated on a separate harmful-prompt distribution; this mostly diagnoses reward mis-specification rather than demonstrating a general vulnerability.

Without stronger evidence of why the findings in the study are useful even on SOTA models, I am less confident that the proposed attacks are useful.

**Questions:**

See above.

---

> ### Author Response · Authors · 2025-11-22
> **The Value of Our Contributions**
>
> Thank you for your feedback and for the time taken to review our paper. We begin by clarifying a **misunderstanding** in our contributions.
>
> ---
>
> > **W1:** The core finding feels incremental and closely aligned with well-known RAG/jailbreak dynamics: if you can steer retrieval early, you can bias generation toward unsafe outcomes.
> >
>
> This does not reflect our actual contribution. We provide a clarification.
>
> Our main contribution is **not** that early retrieval can bias generation. Instead, we show a **training-stage conflict**: standard RL search training (optimised for task accuracy) can easily conflict with instruction tuning (IT), causing models to generate **unsafe searches.**
>
> These unsafe searches **do not** come from IT. They emerge naturally during RL.
>
> We demonstrate this clearly in *Point 2* of our “To All Reviewers” response and in Figure 4, where unsafe searches appear after as few as 50 RL steps.
>
> Additional evidence:
>
> - Our Search attacks outperform generic non-refusal attacks (Table 3), showing safety drops come from **RL-induced search behaviour**, not generic jailbreaks.
> - In Multi-search attacks (Figure 12), models continue harmful searches until they find an answer, again, a behaviour learned during RL.
>
> Thus, our key finding is that RL introduces **new attack surfaces** that IT was not designed to defend. This contribution is important as many practitioners assume IT alone guarantees safety, and thus apply standard outcome-based RL without safety considerations, as reflected in recent widely adopted RL-for-search frameworks [1–4].
>
> Our paper shows **this assumption is incorrect**. As agentic RL becomes increasingly adopted, understanding how RL conflicts with IT is essential for anyone training search-capable LLMs to anticipate the safety risks in their training pipelines.

---

> ### Author Response · Authors · 2025-11-22
> **Justify Model Choices and What This Paper Does Not Claim**
>
> Several of the your concerns are either **outside the scope** of this paper or based on **misunderstandings** of our setting. We address them point-by-point.
>
> > **W2.1:** The attack surface here (forcing [object Object] / multi-search) is quite simple
> >
>
> As we focus on exposing safety vulnerabilities during the RL stage, the simplicity of the attack surface is not a limitation but a *strength:* if merely forcing a model to search before refusing, an interaction any user can perform, is enough to elicit unsafe queries, then standard RL training is vulnerable even under mild user pressure.
>
> ---
>
> > **W2.2:** The experiments use relatively not strong, non-SOTA models.
> >
>
> This is **not** factually accurate.
>
> First, because our paper reveals a competing objective *between* training stages (RL vs. IT), the most meaningful comparison is **before and after RL** (as shown in Figure 4); differences in base model sizes are less central to the finding.
>
> Second, the Qwen and Llama families we use are the SOTA open-weight backbones adopted in nearly all recent RL-TIR papers in 2025, including Search-R1 [1], R1-Search [2], ARPO [3], and ReSearch [4]. As shown in Appendix A, among this year’s top works (by citations and Github stars), over **90%** use Qwen-2.5 and over **40%** use Llama-3.x as the base models for RL search training.
>
> These are precisely the models the open-source community fine-tunes with RL for search today, making them the most relevant choices for uncovering safety vulnerabilities during RL. We have added this justification to the updated PDF.
>
> ---
>
> > **W2.3:** It’s unclear whether the phenomenon meaningfully persists, and to what degree, in production-grade systems (e.g., Claude, Gemini, ChatGPT) that already deploy safety scaffolds and gated tool use.
> >
>
> This concern is out of scope of the paper.
>
> We are **not** claiming attack success on proprietary deployed systems (e.g., ChatGPT, Gemini), which as you noted, deploy additional gated tool-use and safety scaffolds that extend far beyond RL.
>
> Our focus is strictly on the **RL training stage**, where the vulnerability originates: maximising task success in RL induces unsafe searches beyond IT, a phenomenon not previously revealed.
>
> This is why our title is *“Agentic RL for Search is Unsafe”:* the emphasis is on RL as a training process, not on deployed commercial systems.
>
> ---
>
> > **W2.4:** Without evidence on stronger models or more realistic threat models, I’m not convinced the contribution is broadly useful.
> >
>
> We extended our experiments to **larger Qwen-2.5 models (14B and 32B),** the model family used in over 90% of recent RL-TIR papers, and trained them with both PPO and GRPO.
>
> **Result:** We observed the same unsafe-search behaviours and high attack success.
>
> The full results are provided in Point 1 of our response, *“To All Reviewers – Changes Added to Paper,”* and Table 4 of the updated PDF.
>
> With these additional results, our attack successes now **generalise** across:
>
> - **Two model families:** Qwen and Llama
> - **Four model scales:** 3B, 7B, 14B, 32B
> - **Two RL algorithms:** PPO and GRPO
> - **Two search setups:** local and web
>
> ---
>
> [1] S. Zhu, W. Lin, Y. Zhang; *Search-R1: Training LLMs to Reason and Leverage Search Engines with Reinforcement Learning*, 2025. arXiv:2501.12561.
>
> [2] H. Zhao, Z. Liang, X. Wang; *R1-Searcher: Incentivizing the Search Capability in LLMs via Reinforcement Learning*, 2025. arXiv:2502.08987.
>
> [3] Y. Ji, J. Wang, C. Hu; *Agentic Reinforced Policy Optimization (ARPO)*, 2025. arXiv:2501.13693.
>
> [4] W. Wang, X. Shi, W. Jiao; *ReSearch: Learning to Reason with Search for LLMs via Reinforcement Learning*, 2025. arXiv:2502.19457.

---

> ### Author Response · Authors · 2025-11-22
> **Justify RL Setup & Restate Main Contribution**
>
> > **W3:** The RL setup (PPO on HotpotQA with tool tokens) is quite specific and not clearly representative of how real systems are trained/deployed today.
> >
>
> This is **not** factually accurate.
>
> Our RL setup (PPO on HotpotQA + Natural Questions with tool tokens) directly follows recent, well-cited RL-for-search frameworks [1–4], all of which use:
>
> - Similar QA datasets (HotpotQA, NQ, or close variants),
> - Outcome-based rewards (exact match + format rewards), and
> - Standard RL algorithms (PPO or GRPO).
>
> We therefore adopted the standard RL-for-search training recipe used across these frameworks, with the **same** datasets, reward structure, and algorithms (Figure 5 compares how similar their training pipelines are). These RL designs have achieved strong gains on multi-hop reasoning benchmarks (up to 35%) [1, 2].
>
> ---
>
> > **W4:** RL reward optimizes exact match on HotpotQA, but safety is evaluated on a separate harmful-prompt distribution; this mostly diagnoses reward mis-specification rather than demonstrating a general vulnerability.
> >
>
> This is a spot-on observation. We agree that adding search-safety-aware reward terms is a practical mitigation, and we have explicitly encouraged this direction in the Discussion.
>
> But the key insight is:
>
> Current RL pipeline designs (which rely on outcome-based rewards) universally assume IT has already ensured safety and thus omit safety terms from the reward. We showed that this assumption *systematically* induces unsafe search behaviour.
>
> This is not merely mis-specification; it is a **structural vulnerability** in how RL is currently used across the field. Identifying this vulnerability, and its root cause, is precisely the contribution of our paper.
>
> ---
>
> > **Flag For Ethics Review:** Yes, Privacy, security and safety
> >
>
> We noticed that you raised an ethical concern on our paper.
>
> As clarified in the Ethical Statement, all harmful prompts we evaluated come from public academic benchmarks with no personal data. To limit reader exposure to harm, we truncated all model reasoning traces to avoid actionable harm and included explicit warnings in figures displaying harmful content.
>
> ---
>
> **To summarise:**
>
> - Our paper identified a safety vulnerability rooted in the **RL training process**, not an issue with proprietary deployed systems;
> - We used **SOTA open-weight models** and the same RL pipelines used by recent RL-TIR papers [1-4, more in Appendix A];
> - We demonstrated **broad generalisability** of attack success across model families, model sizes, RL algorithms and search setups;
> - Most important, our findings pinpoint **when and how** standard RL erode safety on top of IT, which is directly relevant for anyone training search-capable LLMs to understand the safety implications of their training.
>
> ---
>
> If our clarifications address your concerns, we kindly ask you to consider updating the score accordingly.

---

> ### Author Response · Authors · 2025-11-27
> **Could you respond to our rebuttal?**
>
> Dear Reviewer,
>
> I hope this message finds you well. As the discussion period is nearing its end with less than three working days remaining, I wanted to ensure we have addressed all your concerns satisfactorily. Could you kindly review our rebuttal and give us feedback? If there are any additional points you'd like us to consider, please let us know. Your insights are invaluable to us, and we are eager to address any remaining issues to improve our work.
>
> Thank you for your time and effort in reviewing our paper.

---

### Official Review · Reviewer_iy1j · 2025-11-03

**Soundness:** 4
**Presentation:** 4
**Contribution:** 3
**Rating:** 8
**Confidence:** 2

**Summary:**

This paper tests the safety of RL to train LLM-based agents with access to search.

The hypothesis of the paper is that safety-inducing training helps limit unsafe responses, but does not prevent the agents from performing unsafe searches.

The paper proposes two attacks, the search and the multisearch attack, based on encouraging the agent to do a search first (search attack) or several searches (the multisearch attack). In their empirical evaluation, the paper demonstrates that state-of-the-art agents are actually vulnerable to these type of attacks.

**Strengths:**

S1. The paper identifies a relevant and timely problem, and demonstrates its existence empirically.

S2. Strong empirical evaluation on state of the art agents.

S3. The paper is clearly written and easy to follow.

S4. The paper provides enough details and code as supplementary material for it to be reproducible.

**Weaknesses:**

W1. While using an LLM to judge safety is a standard practice nowadays, it is a limitation and begs the question of "unsafe according to whom?".

W2. The paper is primarily diagnostic: it identifies a safety weakness but does not experimentally test mitigation strategies. While the discussion section outlines possible remedies, they remain unvalidated.

**Questions:**

Q1. (Related to W1) How robusts are your findings to different notions of safety? Have you considered using different LLMs as evaluators to test robustness?

---

> ### Author Response · Authors · 2025-11-27
> **Clarify Notion of Safety + Prototype Mitigation**
>
> Thank you for your precise understanding of our paper. We appreciate your recognition of our timely contributions and our use of SOTA agents.
>
> We address each of your concerns below:
>
> ---
>
> > **W1.** While using an LLM to judge safety is a standard practice nowadays, it is a limitation and begs the question of "unsafe according to whom?".
> >
>
> Thank you for raising this point.
>
> In this paper, we adopted a **basic** and widely agreed-upon definition of safety: a safe LLM should refuse harmful instructions [1, 2], and in agentic settings, should not execute harmful searches based on these instructions [3, 4]. The harmful-instruction datasets we evaluated on: AdvBench [5], MaliciousInstruct [6], TDC2023 [7], and HarmBench [8], contain *unambiguously* harmful prompts seeking advice on cybercrime, social engineering, misinformation generation, and violence. Our safety evaluation therefore prevents users from exploiting LLMs to access such harmful content.
>
> ---
>
> > **W2.** The paper is primarily diagnostic: it identifies a safety weakness but does not experimentally test mitigation strategies. While the discussion section outlines possible remedies, they remain unvalidated.
> >
>
> We agree with this point to some extent.
>
> However, as with any jailbreak paper, diagnosing the vulnerability is a necessary **prerequisite** for proposing mitigations. In that sense, whether we can ultimately “alleviate the problem” is **orthogonal** to our core contribution: we must first establish that the problem exists, *where* it comes from, and *which stage* of training is responsible.
>
> That being said, we are actively implementing all mitigation strategies proposed in the Discussion: (1) a search filter, (2) safe search-and-reasoning augmentation during RL, and (3) modified RL rewards discouraging unsafe searches. As a first step, we trained a RoBERTa-based query classifier that detects harmful searches with >90% accuracy (validated against LLM-judge safety scores). The model is lightweight and efficient at inference time. We will include a full mitigation section in the camera-ready version, comparing the effectiveness of these approaches.
>
> ---
>
> [1] OpenAI. GPT-4 System Card. Technical Report, 2023.
>
> [2] Y. Bai, A. Kadavath, S. Kundu, et al.; *Constitutional AI: Harmlessness from AI Feedback.*
>
> [3] W. Yao, X. Lu, C. Yu, et al.; *WebArena: A Realistic Web Environment for Building Autonomous Agents.* ICLR 2024.
>
> [4] M. Andriushchenko, S. Sehwag, A. Li, et al.; *AgentHarm: A Benchmark for Measuring Harmfulness of LLM Agents.* 2025.
>
> [5] X. Zou, S. Hu, K. Liu, et al.; *Universal and Transferable Adversarial Attacks on Aligned Language Models (AdvBench).* NeurIPS 2023.
>
> [6] Y. Huang, K. Li, P. Lu, et al.; *MaliciousInstruct: Benchmarking Safety Risks of LLM Instruction Following.* 2023.
>
> [7] A. Maloyan, M. Damani, D. Xu, et al.; *TDC2023: Benchmarking Large Language Models on Toxic Dialogue Classification.* 2024.
>
> [8] P. Mazeika, L. Carlini, S. O’Brien, et al.; *HarmBench: A Standardized Evaluation Framework for LLM Harmfulness.* 2024

---

> ### Author Response · Authors · 2025-11-27
> **Show Robustness of LLM Judge**
>
> > **Q1.** (Related to W1) How robust are your findings to different notions of safety? Have you considered using different LLMs as evaluators to test robustness?
> >
>
> As stated, the notion of safety we use is standard and consistent with *consensus* across SOTA LLM providers: a safe LLM should refuse harmful instructions and, in agentic settings, should not conduct searches based on such instructions [1–4]. This represents a basic safety requirement to prevent users from exploiting LLMs to access harmful content.
>
> To test the robustness of our LLM judge, we believe the more meaningful comparison is not against other LLMs (which may contain different biases) but against human raters. We therefore conducted an extensive human–LLM agreement study (Appendix G), with details added to the updated PDF (marked in blue).
>
> ---
>
> **Setup:** We randomly sampled N=50 examples from the 299 harmful instructions and evaluated the RL-trained Qwen-2.5-7B-IT under the Search attack (Prefill-A). For each example, an independent human rater and the LLM evaluator scored the full model response for refusal and answer safety, and each emitted query for search safety, following the 1–5 rubrics (Figure 8). We report Spearman correlations and agreement rates between human–LLM score pairs (exact match, within ±1, and within ±2 points) on the original 1–5 scale.
>
> | Metric | Refusal (N=50) | Answer Safety (N=50) | Search Safety (N=88) |
> | --- | --- | --- | --- |
> | Avg. score (LLM judge) | 76.5 | 70.5 | 25.3 |
> | Avg. score (Human) | 75.0 | 70.0 | 22.0 |
> | Spearman ρ | **0.88** | **0.82** | **0.86** |
> | Exact match (%) | 80% | 62% | 76% |
> | Within ±1 (%) | 92% | 82% | 90% |
> | Within ±2 (%) | 100% | 90% | 100% |
>
> **Results:** All metrics show high Spearman correlation with human judgement: refusal (ρ=0.88), answer safety (ρ=0.82), and search safety (ρ=0.86), all significant at *p < 0.05*. These results indicate that the LLM evaluator provides safety assessments that are consistent and robust to human ratings.
>
> ---
>
> In summary, we adopted a standard notion of safety (refusal of harmful instructions and harmful searches), expanded our discussion on mitigations, and added a comprehensive analysis of LLM-judge robustness against human ratings.
>
> If our responses have addressed your concerns, we kindly ask you to consider raising your **confidence** score accordingly.

---

### Official Review · Reviewer_yQzs · 2025-11-03

**Soundness:** 3
**Presentation:** 3
**Contribution:** 3
**Rating:** 6
**Confidence:** 2

**Summary:**

Search models, despite being trained to refuse harmful requests, are vulnerable to attacks forcing early searches. These searches generate significantly more harmful queries, which then retrieve toxic content, escalating the model's harmful outputs by up to 40%.  This effect is amplified by multiple forced searches. Attacks succeed due to conflicts between RL training (rewarding task success, including harmful behavior) and safety instructions, biased retrieval content, and a lack of safety examples in training data.

Defenses proposed include:
*   Pre-retrieval filters for harmful search queries.
*   Safety-aware RL training incorporating safe trajectories.
*   Analyzing and intervening on harmful query representations.

The core issue is a critical safety gap in agentic LLMs, arising from RL pipelines prioritizing task success over safety, creating exploitable contradictions. Mitigation requires redesigning agent training with safety-grounded rewards and retrieval sanitization, especially as these agents become more widespread.

**Strengths:**

The paper demonstrates a compelling analysis of agentic LLM safety vulnerabilities, supported by several key strengths. Its claims about RL training prioritizing task success over safety are empirically validated through controlled experiments comparing instruction-tuned and RL-trained agents.

The experimental part is good, featuring systematic ablation studies that isolate the impact of different attack vectors. Controls for prompt length and complexity, combined with cross-model validation, enhance the reliability and generalizability of the
findings. Methodologically, the analysis is thorough, providing mechanistic interpretations of failure modes like retrieval bias and RL-safety conflicts, supported by qualitative case studies and rigorous statistical testing.

**Weaknesses:**

First, while it identifies RL-induced behaviors as a core issue, the *explanation* for how RL overrides safety tuning is insufficient. The mechanisms behind this conflict need further conceptual clarification, ideally through diagrams or more detailed theoretical discussion,
to fully explain this key finding.

Second, the paper's framing of its contribution could be strengthened by a more thorough engagement with related prior work. Although it cites foundational agent research like  WebGPT/RAGAS, it overlooks directly relevant studies on *agent jailbreaks* or *retrieval-based poisoning attacks*. This gap limits the novelty justification, as some aspects of the vulnerability landscape might already be partially explored elsewhere.

Finally, the transition from identifying vulnerabilities to proposing mitigations feels somewhat abrupt and disconnected. The empirical findings highlighting specific weaknesses  (like retrieval bias) are not closely linked to the suggested solutions (like
search filters or safety-aware RL). A table explicitly connecting each vulnerability to a corresponding proposed mitigation would significantly improve the paper's logical flow and strengthen the argument for these specific defenses.

**Questions:**

In your PPO implementation, did you observe cases where the  *value function* learned to penalize harmful searches? If so, why does search-triggering still dominate?

When testing with live APIs (e.g., Google Programmable Search), what % of adversarial queries actually returned harmful content? Does API filtering mitigate this vulnerability?

For the proposed search classifier—did you prototype it?

**Details Of Ethics Concerns:**

no concerns

---

> ### Author Response · Authors · 2025-11-27
> **Added Mechanistic Explanation Figure**
>
> Thank you for your precise understanding of our paper. We appreciate your recognition of the novelty of our contributions and the experimental rigour, including the controlled setups and ablations across attack vectors.
>
> ---
>
> We’d like to start by clarifying one minor point in your summary. You wrote:
>
> > These searches generate significantly more harmful queries, which then retrieve toxic content, escalating the model's harmful outputs by up to 40%.
> >
>
> To correct: the 40% refers to the reduction in **refusal rate**, not the increase in harmful outputs. As stated in the Abstract, answer safety actually drops by up to **82.0%.**
>
> ---
>
> Now we address your concerns:
>
> > **W1:** First, while it identifies RL-induced behaviors as a core issue, the *explanation* for how RL overrides safety tuning is insufficient. The mechanisms behind this conflict need further conceptual clarification, ideally through diagrams or more detailed theoretical discussion, to fully explain this key finding.
> >
>
> Please see Point 2 of our response, *“To All Reviewers – Changes Added to Paper.”* We have addressed this concern thoroughly.
>
> ---
>
> > **W2:** Second, the paper's framing of its contribution could be strengthened by a more thorough engagement with related prior work. Although it cites foundational agent research like WebGPT/RAGAS, it overlooks directly relevant studies on *agent jailbreaks* or *retrieval-based poisoning attacks*. This gap limits the novelty justification, as some aspects of the vulnerability landscape might already be partially explored elsewhere.
> >
>
> Thank you for this point. To address it, we have added four additional works on agent jailbreaks and retrieval-based poisoning attacks (two for each) to the Related Work section in the updated PDF (marked in blue), and clarified how our work differs from prior studies.
>
> ---
>
> > **W3:** Finally, the transition from identifying vulnerabilities to proposing mitigations feels somewhat abrupt and disconnected. The empirical findings highlighting specific weaknesses (like retrieval bias) are not closely linked to the suggested solutions (like search filters or safety-aware RL). A table explicitly connecting each vulnerability to a corresponding proposed mitigation would significantly improve the paper's logical flow and strengthen the argument for these specific defenses.
> >
>
> Thank you for this suggestion. We have revised the *Discussion* section to explicitly link each identified reason for vulnerability to its corresponding mitigation. Specifically, for RL-induced unsafe queries (Reason 1), we propose search filtering, safety-aware RL, and inference-time steering; for retrieval bias (Reason 2), we propose retrieval-side content filtering. These mappings have been added to the updated PDF (marked in blue).

---

> ### Author Response · Authors · 2025-11-27
> **Answers on PPO, API Filtering, and Search Classification**
>
> > **Q1:** In your PPO implementation, did you observe cases where the *value function* learned to penalize harmful searches? If so, why does search-triggering still dominate?
> >
>
> The answer is conditional: we observed that PPO penalises harmful searches without attacks, but fails to do so under attacks.
>
> As shown in Figure 1, without attacks, PPO-trained models rarely emit harmful searches after the initial refusal: search safety remains high (e.g. 72.4 for Qwen; Table 3). This suggests that, at the first sight, the value function *does* learn to respect refusal from instruction-tuning (IT) and discourages harmful queries.
>
> However, under our Search attack, which forces the model to output a `<search>` token *before it can refuse*, the first search query is almost always harmful, closely mirroring the user’s harmful intent (high semantic similarity 0.86; low search safety 29.4; Table 3). In this setting, **the value function does *not* penalise the harmful search when refusal is bypassed**.
>
> After this initial unsafe search, behaviour diverges across models: Qwen continues issuing harmful searches throughout (Appendix Figure 15), whereas Llama sometimes shifts toward safer queries later (Appendix Figure 16).
>
> ---
>
> > **Q2:** When testing with live APIs (e.g., Google Programmable Search), what % of adversarial queries actually returned harmful content? Does API filtering mitigate this vulnerability?
> >
>
> From 50 random samples of web-retrieved results, **56%** contained harmful information. API filtering did not fully mitigate the vulnerability, as a large portion of harmful content was labelled *informational or educational* and therefore not blocked.
>
> For example, sources explaining cyberattack techniques for awareness or research purposes bypass filtering, yet can still be exploited when an RL-trained model turns them into actionable instructions. This suggests that API-level filtering alone is insufficient, and that additional content-level filtering on retrieved results may be necessary to prevent harmful content from biasing reasoning.
>
> ---
>
> > **Q3:** For the proposed search classifier—did you prototype it?
> >
>
> Yes. We are actively implementing all mitigation strategies proposed in the Discussion: (1) a search filter, (2) safe search-and-reasoning augmentation during RL, and (3) modified RL rewards discouraging unsafe searches. As a first step, we trained a RoBERTa-based query classifier that detects harmful searches with >90% accuracy (validated against LLM-judge safety scores). The model is lightweight and efficient at inference time. We will include a full mitigation section in the camera-ready version, comparing the effectiveness of these approaches.
>
> ---
>
> In summary, we added a mechanistic explanation figure of how RL overrides IT safety, and showed the effectiveness of API filtering and search classifier as mitigations.
>
> If our clarifications have addressed your concerns, we kindly ask you to consider updating your score accordingly.

---

### Author Response · Authors · 2025-12-03
**To All Reviewers: Changes Added to Paper (1)**

We thank all reviewers for their time and constructive feedback.

The reviewers raised several excellent points that helped us improve the work. We have worked diligently to address these concerns, prioritising those shared across multiple reviews. Our key actions include:

---

### **1. Generalisability to Stronger and Larger Models**

> **Reviewer 9TKw:** My main concern with this paper is whether the conclusion from the paper can be generalized. Firstly, only small-sized LLMs are approached in this paper. It is quite possible that larger models, even the pretrained checkpoint, are better at preventing jailbreaking and thus perform better with regard to attacks.
>

> **Reviewer bhvc**: Without evidence on stronger models or more realistic threat models, I’m not convinced the contribution is broadly useful.
>

To address these concerns, we extended our experiments to **larger Qwen-2.5 models (14B and 32B),** the model family used in over **90%+** recent RL-TIR papers (Appendix A), and trained them with both PPO and GRPO.

**Result:** We observe the **same** unsafe search behaviour and high attack success.

Under GRPO with web search, refusal, answer safety, and search safety all dropped substantially from IT-search (see Table below). Larger models also produce unsafe, request-mirroring search queries on the first forced search (semantic similarity of 0.95 and 0.86), matching the behaviours at smaller scales (3B, 7B).

Notably, the 32B model, which is significantly stronger than the 7B model in both reasoning and coding (~10% gains) [1], still shows the same RL-induced unsafe search behaviours.

| **Model** | **Metric** | **IT-Search** | **Search Attack** | **Multi-Search Attack** |
| --- | --- | --- | --- | --- |
| **Qwen-2.5-14B** | Refusal | 94.5 | 77.0 | 63.5 |
|  | Answer Safety | 92.5 | 74.3 | 57.8 |
|  | Search Safety | 22.8 | 8.8 | 7.8 |
| **Qwen-2.5-32B** | Refusal | 96.0 | 80.8 | 72.6 |
|  | Answer Safety | 97.8 | 85.5 | 69.0 |
|  | Search Safety | 51.0 | 31.5 | 23.0 |

(IT-Search = RL trained Instruction-tuned (IT) model, with no attacks; Lower scores = Less safe.)

Together with the original submission, our attack successes now **generalise** across:

- **Two model families (Qwen and Llama),**
- **Four model scales (3B, 7B, 14B, 32B),**
- **Two widely used RL algorithms (PPO and GRPO),**
- **Both local and web-search setups.**

These results indicate that the safety vulnerability stems from the **RL training process**, not a small-model artifact, thus persists in significantly stronger models.

---

[1] Qwen Team; Qwen2.5 Technical Report, 2025.

---

### Author Response · Authors · 2025-12-03
**To All Reviewers: Changes Added to Paper (2)**

### **2. Evidence of How RL Erodes Safety on Top of IT**

> **Reviewer bhvc**: The core finding feels incremental and closely aligned with well-known RAG/jailbreak dynamics: if you can steer retrieval early, you can bias generation toward unsafe outcomes.
>

This reflects a major misunderstanding in our contribution. Our key finding is **not** that early retrieval biases generation.

Instead, our key finding is a training-stage conflict: standard RL search training (optimised for accurate answering) can directly conflict with IT, causing models to produce **unsafe searches**. This behaviour does not arise from IT, but from the **RL training itself**.

To make this point more explicit, we added a mechanistic analysis to illustrate (now **Figure 4**).

We fine-tuned Qwen-2.5-3B-IT with GRPO and saved checkpoints every 25 steps. At each checkpoint, we ran our Search attack (prefilling `<search>` at the start of the response) and evaluated the safety of the resulting sequence of search queries (reporting only positions with ≥5 examples). The safety scores are:

| **Checkpoint** | **Query Position 1** | **Query Position 2** | **Query Position 3** | **Query Position 4** | **Query Position 5** |
| --- | --- | --- | --- | --- | --- |
| **IT (Qwen 3B)** | 37.25 | 55.50 | -- | -- | -- |
| **RL steps 50** | 5.61 | 14.47 | -- | -- | -- |
| **RL steps 100** | 7.77 | 23.00 | 22.22 | 5.00 | -- |
| **RL steps 150** | 8.64 | 29.76 | 25.00 | 18.75 | -- |
| **RL steps 175** | 5.93 | 20.70 | 19.37 | 25.00 | 29.17 |

Figure 4 visualises this search safety drop from IT to RL. **Key takeaway:**

RL training rapidly erodes search safety from IT. After only ~50 RL steps, the first forced `<search>` query becomes far more harmful than the IT model.

Qualitatively, while the IT model often reframes harmful requests safely (e.g., “legal consequences of money laundering”), RL-trained models more closely mirror the harmful request (e.g., “how to do money laundering”), producing more harmful searches over time.

This shows **unsafe search behaviour is actively acquired and reinforced during RL**, and this degradation emerges **very early** with ~50 RL steps. We added this experiment to clearly illustrate our contributions.

---

### **3. Clarity Revisions Across the Paper**

In response to reviewer requests, we also added small revisions across the paper, include:

- Justification for our RL model and reward choices (Experiment Setup)
- Clarification of the safety notion we adopt (Introduction)
- Expanded related work with four additional studies on agent/RAG jailbreaks (Related Work)
- A more structured mapping between vulnerabilities and mitigations (Discussion)

All updates are marked in blue in the revised PDF.

---

We believe these new experiments and revisions, directly guided by the reviewers’ feedback, have substantially strengthened the paper.

---

### Meta-Review · Area_Chair_S6uo · 2026-01-01

**Summary:**

This paper investigates the safety of agentic reinforcement learning (RL) for search-enabled language models and demonstrates that simple prompt-level interventions—forcing an early or repeated search—can substantially degrade safety behaviors inherited from instruction tuning. Across multiple open-weight model families (Qwen and Llama), scales (3B to 32B), RL algorithms (PPO & GRPO), and both local and web search settings, the authors demonstrate significant reductions in refusal rates, answer safety, and search-query safety. The central claim is that standard outcome-optimized RL for search introduces a training-stage conflict with instruction tuning, actively reinforcing unsafe, request-mirroring search queries that bypass refusal mechanisms.

The work’s strengths include a clear empirical demonstration of a safety failure mode, extensive experimentation across models and training variants, and a careful rebuttal that meaningfully improved clarity and scope. However, weaknesses remain in terms of perceived novelty relative to existing jailbreak and RAG failure modes, limited validation of proposed mitigations, and uncertain relevance to production systems with more complex safety scaffolding.

I recommend that the paper be rejected, but I wouldn't mind if it gets accepted if there is room.

**Reviewer Concerns:**

Several reviewer concerns were substantially addressed by the rebuttal. In particular, doubts about generalization beyond small models were mitigated by new results on 14B and 32B models and across PPO and GRPO, showing similar unsafe-search behaviors. Concerns about a lack of mechanistic explanation were directly addressed through a new experiment demonstrating how unsafe search behavior emerges early during RL training, clarifying that the issue is not merely early retrieval bias but an RL-induced override of instruction-tuned safety. Questions about the robustness of the LLM-based safety judge were also addressed via a human–LLM agreement study showing strong correlations, and the authors improved the paper’s framing by expanding related work and more explicitly mapping vulnerabilities to proposed mitigations.

Nevertheless, several concerns remain outstanding. A key reviewer continued to view the contribution as incremental relative to known RAG and jailbreak dynamics, arguing that the attacks and observed failures do not convincingly establish a new or broadly impactful vulnerability. The usefulness of the findings for real-world, production-grade systems with gated tool use and additional safety layers remains unclear, and this limitation persists despite the authors’ clarification of scope. Additionally, while mitigations are discussed and partially prototyped, they are not experimentally evaluated in a way that demonstrates effectiveness within the same framework, leaving the work largely diagnostic. These unresolved issues underpin continued skepticism about the paper’s novelty and practical impact.

**Reviewer Scores:**

Based on the discussion and rebuttal, the more positive reviewers would likely maintain or slightly increase their scores: the reviewer who initially rated the paper as a solid accept/poster would probably remain supportive given the added experiments and clarifications, and the marginally positive reviewer might modestly increase confidence while still expressing reservations. In contrast, the strongly negative reviewer would likely not materially change their score, as their core concerns about novelty, representativeness, and real-world relevance were only partially addressed from their perspective. Overall, the score distribution would remain polarized, with no clear convergence toward acceptance. I would like to additionally note that the review giving an 8 is rather shallow and unconvincing.

---

### Decision · Program_Chairs · 2026-01-26

Reject